# An Update on Status and Conservation of the Przewalski’s Horse (*Equus ferus przewalskii*): Captive Breeding and Reintroduction Projects

**DOI:** 10.3390/ani12223158

**Published:** 2022-11-15

**Authors:** Mardan Aghabey Turghan, Zhigang Jiang, Zhongze Niu

**Affiliations:** 1State Key Laboratory of Oasis and Desert Ecology, Xinjiang Institute of Ecology and Geography, Chinese Academy of Sciences, Urumqi 830011, China; 2Graduate School, University of Chinese Academy of Sciences, Beijing 100049, China; 3Key Laboratory of Animal Ecology and Conservation Biology, Institute of Zoology, Chinese Academy of Sciences, Beijing 100101, China; 4College of Biology and Geography Sciences, Yili Normal University, Yining 835000, China

**Keywords:** Przewalski’s horse, extinct, captive breeding, reintroduction, interaction, conservation

## Abstract

**Simple Summary:**

The Przewalski’s horse (*Equus ferus przewalskii*), the only extant species of wild horse, was extinct in the wild in 1960s. The wild horse has been successfully saved from extinction by captive breeding projects outside the historic range. Although multiple studies were conducted, the main problems such as loss of founder genes, inbreeding depression, hybridization with domestic horses, high morbidity and mortality, and a lack of reliable prevention strategies and treatment limitations of these problems are still unresolved and require further scientific effort. This review aims to increase understanding of the scientific attributes that make the survival of the species possible and how these attributes can be useful for appropriate design of conservation and management strategies oriented to improve the viability of the existing population of the species.

**Abstract:**

This review summarizes studies on Przewalski’s horse since its extinction in the wild in the 1960s, with a focus on the reintroduction projects in Mongolia and China, with current population status. Historical and present distribution, population trends, ecology and habitats, genetics, behaviors, conservation measures, actual and potential threats are also reviewed. Captive breeding and reintroduction projects have already been implemented, but many others are still under considerations. The review may help to understand the complexity of problem and show the directions for effective practice in the future.

## 1. Introduction

Przewalski’s horse (*Equus ferus przewalskii*), also known as Przewalski’s wild horse, the Asian wild horse, Mongolian wild horse, Takhi or Junggar Horses, is classified by the IUCN as Extinct in the Wild (EW) as no Przewalski’s horse has been seen in the wild since 1969, despite efforts to find them in Mongolia or China [1,2,3,4,5,6,7]. Among the seven extant equid species, the Przewalski’s horse is the only true wild horse in the world [4,8,9,10,11,12,13]. As a species the Przewalski’s horse has been successfully saved from extinction by breeding in captivity [14,15,16,17,18,19] based on a carefully managed founder population [20,21,22]. All Przewalski’s horses alive today are descendants of only 12 wild-caught horses and possibly up to four domesticated individuals [6,7,21,22,23,24,25]. These founders reproduced successfully through captive and reintroduction programs worldwide, and as of 2014 there was an estimated free-ranging population of over 1988 Przewalski’s horses [26,27]. At present, 2500 Przewalski’s horses live in about 112 breeding centers and zoos around the world [6,28]. Reintroduction projects of the species started in 1985 and 1992 in China and Mongolia respectively with a goal to return the horses to their former habitat from captive breeding stock [4,6,25,29,30,31].

Despite extensive studies on captive breeding and reintroduction over the half past century, the long term conservation of the Przewalski’s horses still poses many problems [6,7]. This paper presents an overview of the history and current status of Przewalski’s horses. The main problems which need to be overcome in establishing a wild population are identifying a suitable area and selecting appropriate sources of animals for re-wilding as well as effective conservation management strategies. The consequences of interspecific interaction with other species of equids and ungulates in the release area, including hybridization with domestic horses need to be taken into account. 

## 2. Historic Distribution and Extinction

The 55-million-year history of equine phylogeny has been well-documented from the fossil record. The first visual account of Przewalski’s wild horse-type dates back more than 20,000 years ago [32,33]. Guthrie [34] describes the Przewalski’s wild horse as belonging to the four top grazing herbivores of the Pleistocene Mammoth Steppe, along with the steppe bison (*Bison priscus*), the wooly rhino (*Coelodonta antiquitatis*), and the mammoth (*Mammuthus primigenius*). The Przewalski’s horses were regarded to be a typical representative of the Eurasian steppe fauna [9,10,35,36], although its distribution in prehistoric times was difficult to trace [10]. 

The Eurasian steppe, with higher solar radiation and richer soil fertility, was considered to be the ideal habitat of equids [34,37,38]. Consequently, the Przewalski’s horses were widely distributed over central Asia and Western Europe in prehistoric times [39]. However, in recorded history, it has only been found in the Gobi Desert of Xinjiang, China and Mongolia (85–95° E, 44–50° N), where all Przewalski’s horses were captured around the turn of the 19th and 20th centuries [6,33,40]. The last wild horse was also seen in Mongolia in 1969 [10]. 

There was no description of Przewalski’s horses in Linnaeus’s classic studies [41]. The horses remained unknown in the scientific world until John Bell, a Scottish doctor, sighted the horse in his trip to Peking in 1719–1722 [33]. Colonel Nikolai Przewalski, a Russian explorer collected specimens of a wild horse during his expedition in Xinjiang China in 1880 [42]. Poliakov scientifically documented this species as wild horses for the first time, and he named the specimen *Equus przewalskii*. Later the wild horse was taken as a subspecies of *Equus ferus* [3,43,44,45].

The numbers of Przewalski’s horses in the wild appeared to decline dramatically after World War II. The last wild horse was seen in the Dzungarian Gobi in 1969 [4,10,44]. It is likely that Przewalski’s horses were already quite rare in the wild in the first half of the 20th Century, with only occasional sightings of small groups of horses being reported [10]. Investigations have consistently failed to find the wild populations of the species [4,40,45,46]. The horse was extinct in its original range, and was categorized as Extinct in the Wild by the IUCN [6,7,10,47]. 

Almost all endangered species are threatened because their ecological needs are affected negatively by human interference [18]. However, insight into the extinction of the Przewalski’s horses may also be better understood by studying a variety of information on the historic distribution, taxonomy and ethology of the horse and its relatives [19,48]. Factors such as competition with livestock, hunting, capture of foals for zoological collections, climate change, inbreeding and hybridization, predation and infanticide, restricted access to natural water source have been cited for the extinction of Przewalski’s horses in the wild [4,6,9,10,19,37,49,50,51,52,53,54,55]. Another reason might be a series of harsh winters, for example, recorded in 1945, 1948 and 1956 [10,19]. 

## 3. Captive Breeding and Population Trends

Conservation of endangered species in captive breeding (in zoos and other protected areas) so far has saved several species from extinction [56]. The Przewalski’s horse is an example [10,15,19]. Since the discovery of the Przewalski’s horses, there has been interest in the collection of the species, especially in Western Europe [14,15,16]. At first, six Przewalski’s horses were captured in 1899–1890 and transported to Europe [6,57,58]. It is obvious that an increasing demand for captive breeding of the western world was one of the important factors which caused the extinction of the Przewalski’s horse in the wild [6,10,19]. Thirty-four foals had been transported alive to the west before 1930s. Since then, the numbers of horses captured in the wild reduced drastically [19].

After the Second World War the captive Przewalski’s horse population went through another bottleneck, and there remained only 31 horses in western zoos and wild animal parks, of which only nine were capable to breed [4,49], which apparently descended from the wild stock, but had a slight contribution from domestic horses [4,59]. Horses were exchanged between zoos in order to reduce the level of inbreeding since the 1970s [6,10]. More than 1500 Przewalski’s horses in captivity had been recorded in the studbook by the 1990s [4,19,60], and the number reached 5000 in 2012 [61]. At present, the worldwide population of the living Przewalski’s horses has reached nearly 2500 individuals [28]. Among them, approximately 1360 horses live in the wild in China and Mongolia, 900 were distributed in zoos in Europe, and 120 in wildlife parks in the US [6,28,61]. 

## 4. Reintroduction Projects of the Przewalski’s Horse 

Reintroduction projects in China and Mongolia were proved to be successful in the restoration of the Przewalski horses which disappeared from their former habitats [18]. Semi-reserves were created by the European Conservation Project for preparation for returning wild horse to their original habitat [4,6,29,30,50,62,63,64,65,66]. The Hustai National Park in Mongolia has established which is large enough to maintain groups of Przewalski horses throughout all seasons of the year without any supplemental provisioning. As far as possible, those wild horses are kept totally isolated from external influences, except for necessary interventions such as veterinary care [6,66]. All the reintroduction projects begin with an adaption period in semi-reserves [67]. As early as the 1980s, Przewalski’s horses have been released into large enclosed (semi) reserves in Canada, The Netherlands, Germany, France, Holland, England, Hungary, Ukraine, Uzbekistan and China as a pre-adaption phase before their release to the wild [10,18,68]. By 1990, at least four projects in China and Mongolia have reached the stage of adaptation at the release site [4,10,19]. However, there was a poor coordination of these different reintroduction projects [6]. 

### 4.1. Reintroduction Projects in Mongolia 

Reintroduction project of the Przewalski’s horses into Mongolia was initiated by an expert consolation in 1986 [4,6,19]. Two sites were selected for the reintroduction of Przewalski’s horses to the wild in their former range in Mongolia: Takhin Tal (45.3219° N, 93.3905° E), in the Gobi Strictly Protected Area, an International Biosphere Reserve in the south western part of Mongolia where Przewalski’s horses were last seen in the wild; and Hustain Nuruu National Park, a smaller protected area north of the centre of Mongolia, about 130 km west of Ulaan Baatar [4,19]. The two projects were initiated and supported by different organizations: the former was initiated by the Ministry of Nature and Environment of Mongolia and supported by the Christian Oswald Foundation, Germany and the Mongolian government, subsequently being run by the International Takhi Group with the support of various international sponsors. In 1999 the International Takhi Group (ITG) was established to continue and extend this project in accordance with the IUCN reintroduction guidelines [69,70,71]. The latter was mainly initiated and supported by the Dutch Ministry of Development Aid [4,6,62,72,73,74].

The reintroduction site of Przewalski’s horses in Takhin Tal is a typical semi desert area located in the western part of the Gobi National Park [4,6,19,70]. Five adult Przewalski’s horses including two males and three females were transported from Ukraine in 1992, and released to the wild after a period of adaptation in enclosures, from which the first foal was born in 1992. Another transportation of eight individuals from Ukraine was taken place in 1993 [6]. Sixty horses in ten groups were transported, and 25 foals had been born by the new millennium, of which 14 survived [4,6,29]. In June 2000, seven foals had been born in the free-ranging harem group [70,71,73]. 

The reintroduction of Przewalski’s horse in Hustain Nuruu is set within the context of the broader goals of the restoration and protection of biodiversity within the reserve [4,16,73,74,75]. In June 1992, first group of 16 captive born Przewalski’s horses were transported to the Hustai National Park, where they were held in adaptation enclosures in view of a possible future reintroduction [74]. A second group of 16 Przewalski’s horses was sent in July 1994 to the same site [4,6,76]. In total, 89 Przewalski’s horses were transported to the park in the next eight years [69,70,71,72,77,78]. In 1997 the first harem group was released from the adaptation enclosures, and in 1999 the first foals were successfully raised in the wild [76,77]. The total number of the wild horses in the Hustai National Park reached 122 individuals belonging to nine groups by the new millennium, and became an important vehicle for national park development [71,78]. 

In 2004, 12 captive-born and carefully-selected horses were brought to Khomii Tal, buffer zone of Khar Us Nuur National Park, where they are being held in adaptation enclosures for possible future reintroduction. A second group of horses was shipped in the following year [6,62]. 

The return of the Przewalski’s horses to their native steppes of Mongolia is proven to be successful as the species population has grown steadily in these semi-reserves [17,71,79]. The IUCN Red List of Threatened Species Working Group downgraded the Przewalski’s horse to the status of Endangered in 2011, mainly based on the status of wild horses in Mongolia, for free ranging Przewalski’s horses roaming these sites has reached approximately 350 [29,30]. Today, the number of Przewalski’s horses in the wild in Mongolia has already exceeded 900 [28,80]. 

### 4.2. Reintroduction Projects in China

The current populations of the Przewalski’s horses in China are distributed in three different localities. Among them, the population bred in the wild horse breeding center in Xinjiang Provence is the largest. The other two are the Wuwei–Dunhuang population in Gansu Provence and Beijing–Anxi population in Beijing City respectively [6,7,30,81,82].

#### 4.2.1. The Xinjiang Population 

The reintroduction projects in China has been carried out since 1985, immediately after a conservation action plan for the restoring re-wild population of the Przewalski’s horses was proposed by the Chinese government [4,6,7,30,50]. From 1985 to 2005, 14 males and 10 females captive Przewalski’s horses in five groups from captive facilities in Germany, the United Kingdom, and the United States were transferred to the wild horse breeding center which was established in the semi-desert region of the Junggar Basin in Xinjaing. [6,20,30]. In 2006, a group of six Przewalski’s horses was brought to the center with the aim of improving genetic diversity of the captive individuals [6]. The first foal was born in 1988. Between 1988 and 2013, 339 foals were born and 285 survived at the Centre [50]. In the following year, 11 males and 16 females Przewalski’s horses were released to the wild as a beginning of the restoration program of the species within its former ranges in China [6,30]. Thirteen more groups of Przewalski’s horses have been released since 2013 [6].

In total 89 captive-born Przewalski’s horses (32 males and 57 females) were transported to the Mt. Kalamaili Wild Ungulate Nature Reserve between August 2001 and December 2013 [6]. The first foal was born in 2003 in Mt. Kalamaili Wild Ungulate Nature Reserve by the group released in 2001, and by 2013, a total of 107 foals had been born, of which 88 survived their first year. In 2013, the total population of the released Przewalski’s horses reached 127 individuals divided into 16 groups (13 breeding and 3 bachelor groups) [6,29,50,83]. A preliminary success of restoring the wild population of the Przewalski’s horses has been achieved [30,50,82,83]. The numbers of captive and free-ranging populations in Xinjiang has reached 413 by the end of 2018, but the Przewalski’s horse in China is still listed as Extinct in the Wild because their wild populations still depend on supplemental feeding for their winter survival [6,30,82]. 

#### 4.2.2. The Wuwei–Dunhuang Population 

The distribution sites of the Wuwei–Dunhuang population includes the Wuwei Endangered Animal Breeding Centre and the Dunhuang West Lake Nature Reserve [6]. Since 1990, 18 Przewalski’s horses were brought to the Wuwei Endangered Animal Breeding Centre from the US and Germany as a part of the reintroduction project initiated by the former Ministry of Forestry of the Peoples Republic of China [6,81]. Two releases of wild horses were carried out on 25 September 2010 and 06 September 2012 respectively. Seven and 21 wild horses bred in captivity at the center were kept in enclosures of the west lake national nature reserve to allow them to adapt to the local environment, then they were released. Sixteen foals were born by the re-wildering horses [6]. By the end of 2018, the Wuwei–Dunhuang population increased to 60 as 41 foals were produced. However, the horses were driven into the paddocks to allow for supplemental feeding to increase winter survival, and to reduce competition with domestic horses of local herdsmen [6,81]. 

#### 4.2.3. The Beijing-Anxi Population 

The distribution sites of the Beijing–Anxi population includes the David’s Deer Park in Beijing City and the extreme-arid desert national nature reserve in Anxi County, Gansu Provence [4,6]. Ten Przewalski’s horses brought to the reserve from the United Kingdom in 1985 were kept in the David’s Deer Park for a better acclimatization. In 1997, seven males and three females were translocated to the reserve, which is one of their original areas [6], the David’s Deer Park retained two Przewalski’s horses in the park. By 2018, the number of wild horses in the park increased to five [6], but decreased to one in 2021.

In 1999, the first foal was born in the translocated population in the reserve, and the 2nd, 3rd and 4th generations were born in 2004, 2008 and 2012 respectively. All the individuals introduced in 1997 bred and died before 2012. The number current population of the Beijing–Anxi Przewalski’s horses has reached 23, which still depend on supplemental feeding as the other reintroduced populations do [6]. 

## 5. Research Activities on the Reintroduced Przewalski’s Horse

The Przewalski’s horse was never studied in the wild before its extinction, so only anecdotal accounts of its habitat, genetics, social structure and behaviour before its extinction were available [33]. Nevertheless, all knowledge available on the development of their ethology and ecology in the reintroduction projects is critically important for the understanding of their ecological requirements, as the goal of the research activities on the reintroduced Przewalski’s Horse is to establish a basic reference dataset as well as to provide the scientists with reliable methods for the long-term monitoring of the reintroduction projects worldwide.

### 5.1. Habitat Controversy 

The Przewalski’s horse is a typical grazer within the sub-optimal ranges in arid landscape in central Asia [6,84], as the more favourable steppe region was colonized by nomadic pastoralists over several millennia [19]. The ever-growing human and livestock pressure on the diminishing steppe range forced the Przewalski’s horse to suboptimal ranges, as the last Przewalski’s horse range was located within arid area in the vicinity of water holes, which is not not necessarily its optimal habitat [4,9,19,85]. Of all the wild horse species, the Przewalski’s horse was the one with the most eastern distribution and was most likely well adapted to the arid steppe of the Dzungarian Gobi [64]. However, it is still debated whether these areas represent a mere refuge or were the typical Przewalski’s horse habitat for the living conditions in the Gobi regions are much harsher than in the mountain steppe zone of Hustain Nuuru, where water and forage is more plentiful [4,18,19,49,79,82]. In addition, the adaptation enclosures in the Gobi do not provide enough forage; the wild horses need supplementary feeding year-round [11]. Consequently, the most critical step in the release process is seen in the actual release from the adaptation enclosure [86]. The arid Gobi, in which the Przewalski’s horse can survive as a steppe herbivore, is probably a marginal habitat rather than part of their optimal habitat, and rarity of waterholes in their last refuge should has been considered as a significant factor contributing to their extinction [4,18,19,87,88]. 

The steppe was also considered as optimal habitat of Przewalski’s horse by biological arguments that feral and free ranging horses thrive best on steppe-like grasslands [68,89,90], implying that availability of critical resources may affect the habitat use of the horses [91]. Several studies also suggested that feral horses, including Przewalski’s horses, preferred proximity to rivers, forest and simple plant communities with the preferred species, flatter and lowland areas where they could more easily notice the movement of predators [62,91,92,93]. They use the shade of trees for temperature regulation during summer [94,95], and they climb to windy hillsides located close to such forests to avoid attack from flies [96], as reported previously in feral horses [62,91,92,94,97]. In summary, the view that the Przewalski’s horses are primarily a steppe herbivore has received growing support in the captive breeding as well as reintroduction projects [6,9,11,19,85,86,87,88]. Captive breeding and reintroduction efforts reviewed by Black et al. [88] also indicated that Przewalski’s horses survive and reproduce more successfully when they are dispersed to optimal habitats such grassy semi-desert than they stay at the reintroduction sites which were parts of the arid Gobi. 

### 5.2. Genetic Diversity 

Understanding the genetic relationship between domestic and Przewalski’s horses is critical for formulating conservation and breeding strategies for the species [12,98,99,100,101]. The Przewalski’s horse was recently identified as a descendant of wild horses domesticated in today’s Kazakhstan [98], and significantly different from other horses in morphological traits [1,100,101,102,103]. Genotypical differences also strongly identify the Przewalski’s horse as being more different from other equids [15,104,105,106,107,108,109]. Although genetic studies have yet to identify when and where horse domestication first took place, a recent archaeozoological report indicated the presence of domesticated horses 5500 years ago in Kazakhstan [102,110,111,112]. It is clear from genetic studies that Przewalski’s and domestic horses are also distinct in chromosomal number [101,113,114], but interbreeding between Przewalski’s horses and domestic horses produces fertile offspring [109,110,115,116]. Interspecies embryo transfer has also been successful [23,49,51,117,118].

Przewalski’s horse is the first species to return to its native habitat after living in captivity in small and isolated groups in zoos and parks for generations [4,6,19]. Because of the long and unnatural selection in captivity, it is necessary to map an accurate genetic diversity of Przewalski’s horses to ensure their successful reintroduction to the wild [10,12]. Early studies on the genetics of the species have mainly focused on phylogenetic relationship using a variety of markers, including protein polymorphisms [23,119], chromosomal variation [99,105,108,115,120,121], blood group and allozyme loci [23]. With the widespread use of molecular tools such as mtDNA, immune genes, microsatellite genotyping and genome-wide markers in endangered species [114,118,122,123,124,125,126], more attention has been paid on the evaluation of the level of inbreeding in the Przewalski’s horses, which is critically important for the implementation of long term conservation strategies [12,13,81,101,127,128,129,130]. 

Przewalski’s horses went through large bottlenecks even in pre-historic times [13]. Genetic diversity of the founder population of the Przewalski’s horses was lost significantly due to the genetic drift and bottlenecks of the horse population bred in captivity [4,131]. Only 12 horses have contributed to the gene pool of living wild horse [132], explaining the inbreeding depression in the captive population which results in high mortality, abnormalities in reproduction, and shorter lifespans [15,52,53,54,81,126,128,129,130,133,134,135]. One of the effective ways to minimize inbreeding is to select most distantly related horses to produce offspring [4,15,136], but such a practice has long been debated [137]. Empirical and simulated data [81] showed that the loss of genetic diversity decreases with increasing size of the founder population [81,138,139,140], and this is consistent with the considerable reduction in inbreeding in the reintroduced population in Mt. Kalamaili Wild Ungulate Nature Reserve China since 2001, when the number of the released horses exceeded 100 individuals [81]. The above result is also in consistent with the study of Taylor and Jamieson [141] that differentiation inevitably will occur by genetic drift in small population [142]. 

Another problem is the adaptation to captive environments which impedes the successful implementation of reintroduction [139]. Reducing the number of generations a species spends is considered to be the best approach to minimize genetic adaptation in captivity [143]. Ecological and ethological adaptation mainly happens in the early 5–10 generations, which the Chinese and Mongolian populations have already surpassed [144,145]. It was also suggested that genetic adaptation in captivity poses no threat to the survival of the Przewalski’s horses as they have already spent 16 generations in captivity globally [81].

### 5.3. Social Structure and Ethology 

Social structure in Przewalski’s horses resembles feral horses (*Equus ferus caballus*) in many aspects [4,62,68]. They are both highly socialized within a harem with strong social bounds [146,147,148,149]. There are two types of social groups in Przewalski’s horses: harem groups, including a stallion, mares, and foals, are the breeding units of horses; bachelor groups are the combination of young males [150]. In a harem group, the stallion takes the responsibility of defending females from stallions of other harem groups [151]. Home ranges of harems are commonly separated, but slightly overlapping in some cases [28,30,151,152]. The same phenomenon has also been observed in 2001 in the Pentezug Reserve in Eastern Hungary at the beginning of the reintroduction project with 22 Przewalski’s horses. It is proven that harem groups do not overlap when the home ranges are relatively small [28]. However, overlapping rate became relatively high in 2014 (almost 70%) as the horse population and the number of harem groups continuously grew over the years in the area [28], and this is consistent with the reintroduced horses in China and Mongolia [28,30,151]. Similar social structures and demographic changed were found in domestic horses [68,90] and plain zebras in Africa [152,153]. 

Almost all information on the ecology and ethology of the Przewalski’s horses is from zoos, wild parks and semi-reserves [79], which is important for the understanding of their ecological requirements and ethology. Most scientists assume that the ecology and ethology of the Przewalski’s horses in the wild do not differ greatly from that of feral horses while some predict a more aggressive role of the stallions, based on observations in semi-reserves and zoos [154,155,156]. In captivity, male Przewalski’s horses are more active than females [7] in which solitary males and bachelors move more per hour than harem males [24]. Males devote more time to harem acquisition and defense in the wild, which would maximize male reproductive success while females devote more energy to foraging which influences their reproductive success [24,90,157,158,159]. These gender differences are in consistent with the speculations that, in equids, as in other polygynous mammals, males commonly invest more time and resources in intrasexual competition for access to mates than do females [92] while females typically provide the most parental investment in offspring [160]. 

The type of enclosure also has significant effects on the time budgets of Przewalski’s horse. Horses in small enclosures spend less time stand-resting and change behavior states frequently than horses in larger enclosures [24]. Enclosures with lower density of horses approximate the natural conditions [161], and it is evident that density or space can regulate the conditions of the captive populations of the Przewalski’s horses [162,163]. 

Stress resulting from poor captivity conditions is also known to affect growth, reproduction and resistance to disease in caged animals [143,164] while optimum captivity conditions will result in more natural behavior and greater reproductive success [157]. It was advised that enclosures allow the normal feeding patterns to maintain a healthy herd and to minimize aggressive interactions [154]. Some individual horses may have problems due to lack of experience, lack of acclimatisation or unsuitable physiological or morphological characteristics, and stress associated with increased aggression may have even predisposed colts to disease [79]. In such a case, individuals that are found unable to cope with natural conditions should be identified and excluded in order to protect horses from unnecessary suffering [56]. 

## 6. Conclusions

Przewalski’s horse was extinct in the wild in 1960s, and has been successfully saved from extinction by captive breeding [20,21,22]. Breeding centers for the Przewalski’s horse have been established, and nature reserves have been founded in China and Mongolia [6,30]. With a considerable increase in the number of living horses roaming in the wild, the IUCN Red List of Threatened Species downgraded the Przewalski’s horse to “Endangered” from “Critically Endangered” in 2011 [6,29,30].

Although the Przewalski’s horses have successfully been saved from extinction, they still face actual and potential threats such as habitat deterioration, loss of founder genes, inbreeding, predation by wolves, parasitic diseases, crossbreeding with domestic horses, and so on. We suggest that the real conservation issue for the horses at present, especially in China, is related to intensifying human-wildlife conflicts resulting in increasing human presence and movements in key wildlife habitats along with increasing livestock numbers. We recommend that threats such as over-grazing, road construction and mining activities should be closely monitored to avoid further degradation of the horse’s original habitats.

The long-term threat to the retention of heritable variation in the Przewalski’s horses is loss of founder genes [4]. Further losses of founder genes must be minimized through scientific management. Furthermore, Przewalski’s horses reintroduced into the wild are at risk of crossbreeding with domestic horses. Przewalski’s horses most be isolated during the breeding season [6,15].

Predation by wolves is one of the significant mortality causes in Przewalski’s horses. In the Anxi Nature Reserve, China, for example, 12 wild horses were preyed on by wolves, accounting for more than 30% of the total deaths [165]. Careful monitoring of Przewalski’s horse population dynamics, ecology and ethology in all current and future projects should be carried out to prevent wolf predation and male infanticide in Przewalski’s horses.

Finally, we stress that hybridisation with domestic horses and thus introgression of domestic genes into the reintroduced population is the critical problem to be addressed in the future, in order to secure free-ranging populations of Przewalski’s horses in the wild.

## Data Availability

Not applicable.

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
