# Peer review of "An Update on Status and Conservation of the Przewalski’s Horse (Equus ferus przewalskii): Captive Breeding and Reintroduction Projects"

_animals, 2022, doi:10.3390/ani12223158_

Round 1

Reviewer 1 Report

I read the article with great interest. The idea of compiling information on the reintroduction process is interesting and necessary. As this is a review, it would be good, however, to include the most recent scientific reports on the origin of horses ( f.ex. 10.1038/s41586-021-04018-9 ) The section on Social structure and ethology is too poorly described. Perhaps it should be abandoned and more attention should be paid to publications on genetic research, especially those from the last two  years.

Author Response

Response to Reviewer 1 Comments

I read the article with great interest. The idea of compiling information on the reintroduction process is interesting and necessary. As this is a review, it would be good, however, to include the most recent scientific reports on the origin of horses ( f.ex. 10.1038/s41586-021-04018-9 ) The section on Social structure and ethology is too poorly described. Perhaps it should be abandoned and more attention should be paid to publications on genetic research, especially those from the last two years.

Response:  (in red)

  1. Thank you very much for your great effort and kind attention. The schedule of the manuscript is very tight, and unfortunately i am infected with corona virus and kept isolated at home for weeks of quarantine, it means that i cant access to my office to use my official internet account to download any paper. Thank you very much for your recommendations. I will take your suggestion on the organization of the paper into consideration and do some extra research if the schedule permits, your recommendations on the organization of the manuscript have also give me insight and passion to do the research further in future.
  2. The grammatical errors were corrected one by one and listed in the cover letter.

Thank you very much again

Sincerely yours

Mardan Aghabey Turghan.   

Reviewer 2 Report

This is an excellent comprehensive review about the P Horse and its history. It is well cited and describes the plight of the P horse from the 60's to today. 

There are numerous grammar errors throughout the paper that should be corrected by editing. 

An additional paragraph on breeding successes (discussing hormone manipulation and artificial insemination, etc that has been done in the P horse) would be helpful to allow the reader a more updated history on the conservation of the species. 

Author Response

Response to Reviewer 2 Comments

This is an excellent comprehensive review about the P Horse and its history. It is well cited and describes the plight of the P horse from the 60's to today. 

There are numerous grammar errors throughout the paper that should be corrected by editing. 

An additional paragraph on breeding successes (discussing hormone manipulation and artificial insemination, etc that has been done in the P horse) would be helpful to allow the reader a more updated history on the conservation of the species. 

Response:  (in red)

Thank you very much for your great effort and kind attention, and sorry for the grammatical errors.

  1. The grammatical errors were corrected one by one and listed in the cover letter.
  2. As for you recommendation to work on an additional paragraph on breeding success, i will try if the If the schedule permits. By the way it is pity that i am infected with corona virus and kept isolated at home for weeks of quarantine, it means that i cant access to my office to use myofficial internet account to download any paper. Thanks for the suggestion, i will take it into consideration if there is any possibility for it.

thank you very much again

Sincerely yours

Mardan Aghabey Turghan.  

Reviewer 3 Report

Thank you for putting together a comprehensive summary of the history of these wonderful animals.  There were a number of mistakes in English usage that were very distracting.  I have highlighted many of them in yellow on the attached pdf of your manuscript with comments in the comment box.

Author Response

Response to Reviewer 3 Comments

Thank you for putting together a comprehensive summary of the history of these wonderful animals.  There were a number of mistakes in English usage that were very distracting.  I have highlighted many of them in yellow on the attached pdf of your manuscript with comments in the comment box.

Response:  (in red)

Thank you very much for your kind attention, and i was so touched by your help on the correction of the grammar errors.

The grammatical errors and organization of some sentences were corrected and listed in the cover letter.

Thank you very much again

Sincerely yours

Mardan Aghabey Turghan.  

Round 2

Reviewer 1 Report

I have no more comments. The article, in my opinion, can be published.

Reviewer 2 Report

Appropriate changes were made

Reviewer 3 Report

thank you for the revisions.